# Revealing the Dynamics of Neural Information Processing with Multivariate Information Decomposition

**DOI:** 10.3390/e24070930

**Published:** 2022-07-05

**Authors:** Ehren L. Newman, Thomas F. Varley, Vibin K. Parakkattu, Samantha P. Sherrill, John M. Beggs

**Affiliations:** 1Department of Psychological and Brain Sciences, Indiana University, Bloomington, IN 47405, USA; vparakka@iu.edu; 2Brighton and Sussex Medical School, University of Sussex, Brighton BN1 9RH, UK; spfaber91@gmail.com; 3Department of Physics, Indiana University, Bloomington, IN 47405, USA; jmbeggs@indiana.edu

**Keywords:** higher-order interactions, entropy, information theory, neuroscience, neural recording, computation, cortical circuits

## Abstract

The varied cognitive abilities and rich adaptive behaviors enabled by the animal nervous system are often described in terms of information processing. This framing raises the issue of how biological neural circuits actually process information, and some of the most fundamental outstanding questions in neuroscience center on understanding the mechanisms of neural information processing. Classical information theory has long been understood to be a natural framework within which information processing can be understood, and recent advances in the field of multivariate information theory offer new insights into the structure of computation in complex systems. In this review, we provide an introduction to the conceptual and practical issues associated with using multivariate information theory to analyze information processing in neural circuits, as well as discussing recent empirical work in this vein. Specifically, we provide an accessible introduction to the *partial information decomposition* (PID) framework. PID reveals redundant, unique, and synergistic modes by which neurons integrate information from multiple sources. We focus particularly on the *synergistic* mode, which quantifies the “higher-order” information carried in the patterns of multiple inputs and is not reducible to input from any single source. Recent work in a variety of model systems has revealed that synergistic dynamics are ubiquitous in neural circuitry and show reliable structure–function relationships, emerging disproportionately in neuronal rich clubs, downstream of recurrent connectivity, and in the convergence of correlated activity. We draw on the existing literature on higher-order information dynamics in neuronal networks to illustrate the insights that have been gained by taking an information decomposition perspective on neural activity. Finally, we briefly discuss future promising directions for information decomposition approaches to neuroscience, such as work on behaving animals, multi-target generalizations of PID, and time-resolved local analyses.

## 1. Introduction

A grand challenge of modern neuroscience is to discover how brains “process information”. Though there is debate regarding what *information processing* means precisely, it is clear that brains take in sensory signals from the environment and use that information to generate adaptive behavior informed by their surroundings. The ways that sensory information is modified and transformed are poorly understood. It is known, however, that such processes are distributed over integrative interactions in neural circuits throughout the brain and involve large numbers of interacting neurons. We refer to this collective activity as neural information processing.

There are many basic unanswered questions about information processing. What types of circuit connectivity are most favorable for neural information processing? How is neural information processing affected by the overall structure and correlations present in incoming information flows? What are the physiological correlates of information processing at the level of neural circuits? Are distinct transformations promoted by distinct physiological systems? Do distinct transformations relate to the demands of different environmental or behavioral tasks? Modern neuroscientific research has begun to probe these issues. While many questions remain, patterns have begun to emerge, replicated across multiple studies, that suggest that the structure of information processing in the brain follows certain reliable trends and may serve particular purposes with respect to ongoing biological and cognitive processes.

Historically, there have been two key limitations hindering the study of information processing by neural circuits. The first was a lack of appropriate data, since the analysis of synergistic dynamics requires large amounts of data to accurately infer the structure of multivariate dependence. Until recently, such data were difficult to access. In the last decade, however, with innovative new electrical and optical recording technologies that can record from hundreds or thousands of neurons at sub-millisecond temporal resolution, this limitation has abated considerably. The second was a lack of appropriate analytical approaches. Despite the recent explosion of data, a rendering processes is still needed to extract interpretable insights from terabytes of recordings.

In this review, we describe how information theory, and partial information decomposition [1] in particular, has been successfully applied to achieve partial resolution of the second limitation. This is not intended to be an exhaustive review of relevant findings but rather to serve as an approachable overview for those inside and outside neuroscience who may be interested in how multivariate information-theoretic analysis can be applied to questions of interest to them. For those already familiar with neuroscience and information theory, we take this opportunity to highlight recent advances in our own work and our vision for future directions in this area (e.g., [2,3,4,5]).

## 2. Tracking Information in Neural Circuits

When used to study neural circuit function, information theory analyzes knowledge about the activity of one or more neurons and reduces the uncertainty about the states of other neurons. While the overall intuition is conserved, the practical considerations can vary widely, depending on the type of data being analyzed. The properties of “neural activity” differ, depending upon whether the activity was recorded at the field level and is represented by continuous, real-valued data (e.g., local field potentials as obtained in broadband extracellular recordings, electrocorticography, or electroencephalography) or at the single unit level and is a discrete ON/OFF process (e.g., “spiking” activity recorded electrically with intracellular electrodes, multielectrode arrays, tetrodes, or silicon probes, or optically with endoscopes or two-photon microscopes). Both types of data are common in neuroscience, although they require different approaches in an information-theoretic context (for further discussion, see Section 6).

### 2.1. Defining Key Dimensions

When embarking on an analysis of information processing in neural data, there are several key dimensions that must be defined, as they will inform what kinds of analyses can be performed and what the technical requirements might be. Information theory generally requires one to think in terms of “states” that a variable can adopt and transition between over time. In the case of continuous data such as local field potentials (LFPs), the “states” are continuous (conceivably spanning the set of real numbers), while in the case of spiking data the “states” are discrete and taken from a finite-sized support set. Information theory is generally considered to be “easier” in the case of discrete random variables as opposed to continuous ones, although options do exist for continuous parametric and nonparametric analyses.

Another dimension that must be specified is that of the elements of interest and the proposed model of interaction between them. Typically, this involves neurons or brain regions, and the connections are assumed to be synaptic connections or communication over white matter tracts. However, the elements can also be recording electrodes (if spike-sorting or source-modeling is not an option). The nature of the different elements under study will inform what kinds of inferences can be made about the underlying activity and how it relates to structure, behavior, and cognition.

Finally, there is the question of time. Whether examining field or unit activity, it is important to consider the temporal scale of the analysis. Digitally recorded data are almost always presented as discrete frames, sampled at a fixed rate. The choice of bin size (i.e., the temporal scale) is important for determining the sensitivity of the analysis to transformations at a given scale. For example, analyses of synaptic integration, the putative basis for some portion of circuit information processing, should be resolved at the scale of 1 s to 10 s of ms [6,7], whereas analyses of up–down cortical states should capture the dynamics at 100 s to 1000 s of ms [8]. Indeed, it is important to explore the role of timescale directly by examining a range of scales [3]. For more about time binning, see [9].

### 2.2. Tracking Information

Given two elements (e.g., neurons), one serving as an information source and another as a target receiving information from that source, the tools of information theory make it possible to track how neural activity propagates across a neural system. Information-theoretic tools such as mutual information [10] and transfer entropy [11] are well suited to this purpose.

Mutual information [12] can measure the dependence in the spiking between two neurons:(1)$$I\left(X;Y\right):=\sum_{\begin{array}{c} x\in X\\ y\in Y \end{array}}P\left(x,y\right)\log_{2}\frac{P(x|y)}{P(x)}$$
where P(x,y) is the probability distribution of the joint state of *X* and *Y*, P(x) is the marginal probability of *X*, and P(x|y) is the conditional probability X=x given that Y=y. I(X;Y) measures how our ability to correctly infer the state of *X* changes, depending on whether we are accounting for (potentially) shared dependencies with *Y*. It can be thought of as a nonlinear correlation between two patterns of activity [3,13]. This similarity is related to *functional connectivity* [14]. It does not, however, quantify information *transfer* between the neurons, as it is an undirected measure that describes the instantaneous dependency between two variables and has no notion of the time-directed structure that we intuitively understand as “transfer” or “flow.”

Transfer entropy [11] is well suited to measuring how much the *past* activity of one neuron (e.g., $X_{p}$) accounts for the immediate future activity of another neuron (e.g., $Y_{t+1}$), conditioned on *Y*’s own past ($Y_{p}$):(2)$$\begin{array}{cc} TE & (X\to Y):=\\ \; & \sum_{\begin{array}{c} y_{t+1}\in Y\\ x_{p}\in X_{p}\\ y_{p}\in Y_{p} \end{array}}P\left(y_{t+1},y_{p},x_{p}\right)\log_{2}\frac{P(y_{t+1}|x_{p},y_{p})}{P(y_{t+1}|y_{p})} \end{array}$$

TE(X→Y) is read as the transfer entropy from *X*’s past to *Y*’s future. It is important to note that $X_{p}$ does not necessarily have to be a single moment or bin but can be a multi-dimensional and potentially non-uniform embedding [15,16,17]. TE is understood as quantifying how much the *past* of the source variable reduces our uncertainty about the *future* of the target variable, after accounting for information disclosed by the target variable’s own past (autocorrelation). Whereas mutual information is a measure of *functional connectivity*, transfer entropy measures *effective connectivity* [14]. In other words, transfer entropy provides a measure of *information propagation*. In addition to measuring the magnitude of information flow between two neurons, in the special case of binary signals the TE can be modified to also provide a measure of excitation/inhibition balance using the *sorted local transfer entropy* [18].

## 3. Information Processing in Neural Circuits

Information transfer captures the overall “flow” of information through the system; however, it is limited in its ability to reveal how different streams of information “interact” and how neurons produce novel or modified information from multiple sources [19]. How does one examine *information processing* itself? We propose that one successful avenue is that of multivariate information decomposition [1].

Multivariate information decomposition is a branch of information theory that endeavors to dissect out the atoms of information processing from a system of interacting units. Consider the simplest case of two neurons ($X^{1}$ and $X^{2}$) that jointly synapse onto a target *Y*. To understand how Y processes input from the sources, we are interested in knowing how the *past* states of $X^{1}$ and $X^{2}$ reduce our uncertainty about the *future* state of *Y*. To understand this, we must know:The future state of *Y* ($Y_{t+1}$);The past state of the target *Y* ($Y_{p}$);The past state of $X^{1}$ ($X_{p}^{1}$);The past state of $X^{2}$ ($X_{p}^{2}$).

Information transfer (discussed above) tells us about how the past of either source $X^{i}$ informs the future of *Y*, but to understand “processing”, we are interested in how the *interaction* between the two source variables’ past influences the target’s future in a way that is not reducible to either source considered individually. This parsing is accomplished using the partial information decomposition (PID) framework [20]. In the next section, we cover the premise and operation of the PID framework. The following section illustrates the application of the PID framework with examples from our own work. In the final section we offer practical pointers for how to get started with using PID in new settings.

## 4. PID: Partial Information Decomposition

In this section, we will dive into the details of partial information decomposition, with the aim of providing an accessible introduction to the framework and outlining the common technical concerns that a prospective analyst must address. We begin with a formal statement of the problem and build intuition by examining the special case of two parent neurons connecting to a target neuron. This allows us to introduce the notions of redundant, unique, and synergistic information in an accessible way. We then generalize to the case of an arbitrary number of parents, which requires more advanced mathematical machinery. Finally, we address the problem of defining “redundant information” (a standing issue in the field, but an essential component of any PID analysis).

### 4.1. The Basic Problem and Intuition

Individual neurons receive inputs from from many “parent” neurons, which are “integrated” into a single “decision” by the target neuron, i.e., whether to fire an action potential or not. Exactly how individual neurons “compute” their future behavior as a function of their inputs is a long-standing question in computational and theoretical neurosciences. We can quantify the total amount of information the inputs provide about the decision state of the target neuron using the *joint mutual information*:(3)$$I\left(X;Y\right)=\sum_{x\in X}\sum_{y\in Y}P\left(x,y\right)\log_{2}\frac{P(y|x)}{P(y)}$$
where $X=\{X^{1},\ldots ,X^{N}\}$ is the set of all pre-synaptic parent neurons and *Y* is the single post-synaptic target neuron. We can also compute the individual information that single parents provide about the target with the *marginal mutual information*
$I(X^{i};Y)$. Interestingly, it cannot be assumed that the joint mutual information (the “whole”) is reducible to a sum of all its component marginal “parts”.
(4)$$I\left(X;Y\right)\neq \sum_{i=1}^{\left|X\right|}I\left(X^{i};Y\right).$$

If the left-hand side of Equation (Equation 4) (the “whole”) is *greater* than the right-hand side (the sum of the “parts”), then there is some information about *Y* that is disclosed by the joint state of all the neurons that is *not* disclosed by any individual parent. In this case, the system exhibits *synergistic* dynamics, and the target neuron can be thought of as performing a kind of integrating “computation” on all of its inputs considered jointly (this is also sometimes referred to as “information modification” in the literature [19]). Conversely, if the whole is less than the sum of its parts, then there must be *redundant* information about *Y* instantiated multiple times over the parent neurons that is “double counted” when summing the marginals. The way that these pieces fit together is illustrated in Figure 1.

This inequality has been recognized for decades; however, it was not until the seminal work of Williams and Beer [1] that it was recognized how to algebraically decompose the total joint mutual information into the particular contributions of the parts (and higher-order ensembles of parts). Since its introduction, this framework, termed *partial information decomposition*, has been widely applied in fields from theoretical and computational neuroscience [9,21] to climate modeling [22] and sociology [23].

Here, we focus on recent applications of PID to activity recorded from systems of spiking biological neurons and on how this statistical framework can provide insights into the nature of “computation” in cortical circuits. In the next section, we introduce the intuition behind PID in more detail, before building to the full mathematical definition and discussing the various technical desiderata that an aspiring researcher must consider when applying a PID analysis to their own data.

#### The Special Case of Two Parents

To build intuition, it is instructive to consider the particular case of two parent neurons ($X^{1},X^{2}$) synapsing onto a single target neuron (*Y*). We would like to be able to decompose the joint mutual information between the two parents and the target into an additive set of non-overlapping “atomic” information components:(5)$$\begin{array}{cc} I(X^{1},X^{2};Y)= & \mathrm{Red}(X^{1},X^{2};Y)+\\ \; & \mathrm{Unq}(X^{1};Y)+\\ \; & \mathrm{Unq}(X^{2};Y)+\\ \; & \mathrm{Syn}(X^{1},X^{2};Y) \end{array}$$
where $\mathrm{Red}(X^{1},X^{2};Y)$ is the information about *Y* that could be learned by observing $X^{1}$ or $X^{2}$, $\mathrm{Unq}(X^{i};Y)$ is the information about *Y* that could only be learned by observing $X^{i}$, and $\mathrm{Syn}(X^{1},X^{2};Y)$ is the information about *Y* that can only be learned by observing the joint states of $X^{1}$ and $X^{2}$ together. This is illustrated in Figure 1.

Furthermore, the two marginal mutual information terms can be decomposed accordingly:(6)$$\begin{array}{cc} I(X^{1};Y) & =\mathrm{Red}\left(X^{1},X^{2};Y\right)+\mathrm{Unq}\left(X^{1};Y\right)\\ I(X^{2};Y) & =\mathrm{Red}\left(X^{1},X^{2};Y\right)+\mathrm{Unq}\left(X^{2};Y\right) \end{array}$$

The result is an underdetermined system of linear equations, with three known values (the mutual information terms) and four unknown values (the partial information terms). If any of the unknown terms can be defined, then the other three emerge “for free”. Unfortunately, classical Shannon information theory provides no unique solution to any of these, and so considerable work has been devoted to developing formal measures of each. The most common approach is to define a *redundancy function* such as the original proposal of $I_{\min}$ [1] (for more on redundancy functions, see Section 4.3), although there have also been proposals that start with the unique information [24,25] or synergy [26,27].

By far the most common approach to applying PID in neurosciences involves analyzing only triads, with two parents and a single target (for example, [2,3,4,20,28]); however, the PID framework is conceptually powerful enough to support any number of pre-synaptic elements. Unfortunately, in the case of three or more parents, the resulting system of equations is not as constrained as in the two-element case, as the number of distinct partial information atoms grows much faster than the number of known mutual information terms. The general PID framework requires the introduction of more involved mathematical machinery.

### 4.2. The Partial Information Lattice

The first step in general PID is to define the set of information *sources*, i.e., all subsets of **X** (including **X** itself) that can uniquely disclose information about *Y*. This is equivalent to defining the non-empty power set of **X**. For example, in the case where $X=\{X^{1},X^{2},X^{3}\}$, the source set **S** is given by:(7)$$S=\left\{\;\begin{array}{cc} \; & \left\{X^{1}\right\},\left\{X^{2}\right\},\left\{X^{3}\right\},\\ \; & \left\{X^{1},X^{2}\right\},\left\{X^{1},X^{3}\right\},\left\{X^{2},X^{3}\right\},\\ \; & \left\{X^{1},X^{2},X^{3}\right\} \end{array}\right\}$$

One of the key insights from Williams and Beer was that given a suitable measure of redundancy (which we will refer to as $I_{\cap }\left(\cdot \right)$), it is possible to decompose the total joint mutual information into a finite set of unique, partial information atoms based on how all the elements of **S** share information about *Y*. This idealized redundancy function must satisfy a number of axioms (for further discussion, see [1,29,30]). For our purposes, it is sufficient to assume that an idealized $I_{\cap }\left(\cdot \right)$ function exists.

The domain of $I_{\cap }\left(\cdot \right)$ is given by the set of all combinations of **S** such that, in a given collection of sources, no individual source is a subset of any other. We define this set as A:(8)$$A=\{\alpha \in P_{1}\left(S\right):\forall A_{i},A_{j}\in \alpha ,A_{i}⊄A_{j}\}$$

Here, $P_{1}$ is once again the non-empty power set (for a discussion of the no-subsets requirement, see [1,29,30]). Intuitively, this can be understood by analogy with a Venn diagram. If one circle is completely subsumed by another circle, the redundant area shared between them is just that of the smaller circle. This restriction means that A is also partially ordered into a lattice (which we called the *partial information lattice*):(9)∀α,β∈A,α⪯β⇔∀B∈β,∃A∈αs.t.A⊆B

This can be read as indicating that one element of A (α) is said to precede another element (β) on the lattice if every element of β has an associated element in α that is a superset or equal to it. Given this partially ordered lattice structure and our redundancy function $I_{\cap }\left(\cdot \right)$, it is possible to recursively quantify the value of every partial information atom (Π(α)) via Mobius inversion:(10)$$\Pi \left(\alpha ;Y\right)=I_{\cap }\left(\alpha ;Y\right)-\sum_{\beta ≺\alpha }\Pi \left(\beta ;Y\right)$$

Intuitively, this can be understood as formalizing the notion that the information about *Y* that is *only* disclosed by the various $A_{i}\in \alpha$ is that information that is not accessible in any simpler combination of sources lower down on the lattice. Representative example lattices for two and three predictor variables are shown in Figure 2. Finally, we can confirm that we have completed the decomposition by observing that:(11)$$I\left(X;Y\right)=\sum_{i=1}^{\left|A\right|}\Pi \left(\alpha_{i};Y\right)$$

### 4.3. Choosing a Redundancy Measure

Applying PID analyses to real data requires some notion of redundant information ($I_{\cap }\left(\cdot \right)$) to be operationalized. To date, no single universally accepted measure has been proposed. All measures have their own trade-offs and drawbacks (such as only being defined for systems of a fixed size, only being amenable to discrete random variables, or requiring arbitrary thresholds). There are, at present, close to a dozen competing redundancy functions (see [22,29,31,32,33,34,35,36,37,38,39,40]). In the absence of a single accepted measure, different contexts may require the choice of different functions. For example, having more than two predictors precludes the measures proposed in [24,31], while continuous data cannot be analyzed with the measures proposed in [24,40], and so on. The majority of the studies discussed in this paper used the original $I_{\min}$ measure proposed in [1], although this measure has been criticized for unintuitive behavior [29,39]. For a deeper discussion of the practical considerations, see Section 6.

## 5. PID in Action

We next illustrate the application of information decomposition to the *empirical study* of neural information processing in biological circuits. This is not intended to be an exhaustive review of relevant studies. Rather, the goal is to offer a crisp window into the types of insights that can emerge. We draw preferentially on our own work out of convenience, but this work in turn was inspired by, and complements, other exemplary lines of empirical work (for examples, see [5,21,35,41,42,43,44,45,46]).

To gain traction on this problem, we used 512-channel multielectrode arrays to record hundreds of individual neurons in vitro from organotypic cultures derived from mouse cortical slices. We then rendered the single-unit-level functional recording into an effective network, capturing the microcircuit dynamics using transfer entropy. In our recordings, this revealed that about 0.4–1.0% of all possible directed connections between neurons were significant. From these, we then identified all triads consisting of two source neurons connecting to a common target neuron (totalling thousands of triads) and performed the full partial information decomposition into redundant, unique, and synergistic atoms. The numerous triads observable in a single recording make it possible to perform within-recording comparisons of the features that correlate with synergistic integration. This is illustrated in Figure 3.

For example, one can ask whether synergy varies systematically as a function of the amount of information propagation—the transfer entropy from sources to a target—across triads. When we performed this analysis, examining 25 separate recordings at timescales varying from 1 ms to 14 ms, we found that synergy was strongly correlated (ρ≫0.57 for all recordings and timescales; see Figure 4A). Interestingly, this analysis revealed that—for these recordings—the amount of synergy observed in a given triad was reliably about a quarter of the transfer entropy for that triad [2]. The implication of this is that feedforward information propagation between source neurons and a target neuron is a reliable predictor of information processing.

### 5.1. Synergy in Rich Clubs

The observation that information propagation correlates with information processing suggests that rich clubs may be dense cores of information processing. Rich clubs generally represent the set of best-connected nodes of a network that are mutually interconnected with a probability higher than that expected by chance [48]. The *rich club coefficient* quantifies just how much more densely connected a given set of nodes is than would be expected by chance. Rich clubs, in the context of the effective networks built from cortical circuit spiking recordings, are comprised of the neurons that propagate the most information (i.e., sending and receiving). By definition, rich clubs are disproportionately dense in information propagation: 20% of the neurons account for 70% of the information propagation in organotypic cortical cultures [49].

PID allowed us to test whether triads inside the rich club have greater synergy than those outside the rich club [2]. Unambiguously, rich club triads processed more information than triads outside rich clubs. Numerically, the triads inside the rich clubs had 2.7 times the amount of synergy as the triads not in the rich club (Figure 4B). Due to rich clubs having a high density of triads and rich club triads having a greater synergy per triad, a majority of the network-wide synergy was accounted for by rich club triads. Despite the fact that that less than 40% of the neurons were in the rich club, the rich club triads accounted for ∼88% of the network-wide synergy. We also found that there was a strong positive correlation between the rich club coefficient and the average synergy produced by rich club triads, suggesting that the rich club structure itself was related to the amount of computation.

To understand better why cortical rich clubs are so dense in information processing, we examined other determinants of synergistic integration. We asked, for example, how the connectivity motif of computational triads related to synergy. We also asked whether the similarity of the spike trains converging on a target neuron from two source neurons impacts the amount of synergistic integration. Each of these is unpacked next.

### 5.2. The Importance of Feedforward, Feedback, and Recurrent Connections

Understanding how connectivity patterns relate to synergistic integration may be useful for predicting how a network with a given topology might perform computationally. Connectivity motifs characterize and describe the different ways a fixed number of nodes can be connected. For example, in a three-element triad, the connectivity can be parsed into feedforward, feedback, and recurrent types. Feedforward connections are those from the source neurons to the target neuron. Of the two possible feedforward connections from source neurons to a target, both must exist in any *computational triad*. Feedback connections are those from the target back to the source neurons. There may be zero, one, or two of these in a computational triad. Recurrent connections are connections between the source neurons. Again, there may be zero, one, or two recurrent connections. The question for our analysis was whether synergistic integration was sensitive to the number of feedback or recurrent connections.

Using PID, we found that triads with more recurrent connections also had greater synergy [4]. Feedback connections were less predictive of synergy but tended to correlate with reduced synergy. Numerically, in comparison to the simplest computational triads (those with no feedback or recurrent connections), triads with two recurrent connections and no feedback connections had 50% more synergy (Figure 4C). Triads with two feedback connections but no recurrent connections had 10% less synergy than the simplest computational triads. Modeling the synergy based on feedforward, feedback, and recurrent connection *strengths* instead of the *numbers* of connections, using multiple linear regression, revealed a similar pattern (Figure 4). The feedforward connections were found to be positively correlated with synergy and accounted for the most variance. The recurrent connections also were positively correlated, though they accounted for a smaller amount of the variance than the feedforward beta weights. This may be related to another prior finding that synergy is greater downstream of neurons that propagate information to many target neurons [28]. Finally, feedback connections were not significantly related to synergy. The finding that greater synergy is found in triads with greater connectivity between source neurons offers some perspective into why rich club triads are synergy-dense. Not only are rich club triads likely to have strong feedforward connections, they are also likely to have additional connections, including recurrent connections, that add to the overall synergy.

### 5.3. Synergy from the Convergence of Correlated Information

In addition to questions about network topology, we also examined whether synergy varies based on the functional similarity of converging information streams. We tested whether there was a systematic correlation between the synergy of a triad and the mutual information in the source-neuron activities. Initially, we tested this for timescales in the synaptic range (<14 ms), consistent with the analyses described above. The result was unambiguous. The greater the mutual information between source neurons at the synaptic timescale, the greater the synergy [3]. More information processing occurs where similar information streams converge (Figure 4E, the three leftmost panels for synaptic timescales). To test whether synergy increases indefinitely as mutual information grows at longer timescales, we explored a range extending well past the synaptic range (time bins up to 2.25 s wide). Interestingly, synergy only increased up to a point. The peak in synergy occurred when the mutual information was about 7% of the maximal value, regardless of the timescale. Past this level, the synergy began to decrease (this level is marked by the vertical dotted line in Figure 4E). In the context of explaining the density of synergy in the rich club, the strong positive relationship between synergy and mutual information at synaptic timescales suggested that the activity of rich club neurons is generally more correlated than the activity of neurons outside the rich club. Indeed, rich clubs consist of a disproportionate number of inhibitory neurons with correlated spiking activity that synergistically predict the dynamics of the rest of the network [50].

Leveraging the multiple outputs that PID offers, including redundancy as well as synergy, we were able to explore how the overall composition of information propagation varies across timescales [3]. As we lengthened the timescale, the total multivariate transfer entropy between the source neurons and the target neuron increased steadily across all timescales examined. As noted already, however, synergy increased only at the shorter timescales and decreased at the longer timescales. Observing the redundancy offered an explanation. Redundancy was positively related to mutual information at all timescales. Indeed, the slope relating the redundancy to mutual information became steeper at longer timescales (Figure 4E). This suggests that as the similarity of the converging information grows past some point, it becomes redundant, and the total amount of synergistic output is suppressed.

### 5.4. Caveats to Consider

There are a few important caveats to the empirical work described here that warrant caution and further attention. The first is that the functional networks from which triads were drawn and against which the synergy values were compared were generated from bivariate transfer entropy analyses. This matters because it has been shown that bivariate analyses overestimate the significance of bivariate edges [51]. A second caveat, related to the source of the data rather than the analyses, is that the processing we studied was in cortical cultures. This matters because it leaves open key questions about what we would observe if these analyses were performed in vivo. While organotypic cultures have been proven to share many of the key properties of uncultured cortical circuits (i.e., in the brain studied in vivo or via ex vivo histological analysis) [47], the demands placed on a circuit in the context of a behaving animal are certain to differ from those placed on cultures in vitro.

### 5.5. PID in Behaving Primates

These caveats have informed recent applications of PID to questions of neural dynamics. Varley et al. [5] analyzed spiking neural activity from the pre-motor and motor regions of three macaques while the monkeys were engaged in a multi-phase behavioral task involving symbol recognition, motor planning, memory storage, and motor execution [52]. As with the in vitro work, this study began by inferring transfer entropy networks for each behavioral epoch and then calculated the PID for every triad in the resulting networks. Rather than using a bivariate transfer entropy function, however, the authors used a serial conditioning algorithm to infer the multivariate TE networks [51,53,54] that contextualize every source–target relationship based on the global patterns of information flow. Despite this difference, many aspects of the in vitro results were successfully replicated in animal models, including the presence of a rich club and an increase in synergy in high-degree nodes. The authors also found that, in addition to a positive correlation between synergy and target degree, there was also a positive correlation between the local clustering coefficient and the synergy, which is significant because it points to a relationship between the local computational structure of a single neuron and its local environment (including neurons it may not directly interact with).

The ability to assess how the structure of information processes changes in response to the demands of different tasks is a significant departure from the limitations of in vitro research. The researchers found that while brain activity was generally synergy-dominated (as opposed to redundancy-dominated), during movement execution (a reach-and-grasp action) the relative abundance of redundant information increased dramatically. This increase in relative redundancy was interpreted as a response to the practical requirements of the task: during motor execution, the brain needs to send the “move” signal to distant muscles, and to ensure high-fidelity transmission the brain may duplicate information many times over (minimizing the risk of a single corrupted signal leading to erroneous behavior). Similarly, different brain regions had different relative synergies during different tasks. For example, AIP (a pre-motor region) had the highest synergy during the fixation/cueing epoch, which then dropped off during movement, while M1 (the primary motor region) had the lowest synergy during the memory epoch, which then exploded during movement execution.

These results show not only that synergistic information dynamics is a feature of ongoing, spontaneous neural activity but also that synergy seems to reflect behaviorally specific patterns of dynamical activity in the cortex.

## 6. Practical Considerations in PID

In this last section, we provide a basic orientation to the practicalities of performing analyses using the PID framework. This includes identifying and selecting the correct tools for a given context and the considerations required in interpreting the results. The first major consideration when considering a PID analysis is the form of the data. While the construction of the lattice does not assume discrete or continuous data, different redundancy functions are more or less capable of managing continuous variables. The original measure, $I_{\min}$, as well as several subsequent developments including $I_{BROJA}$ and $I_{sx}$, are only well-defined on discrete variables (although work on continuous generalizations is ongoing (see: [55,56,57])). Other measures such as the minimum mutual information [22,35] and the Gaussian dependency decomposition [58] are exact for Gaussian variables but not necessarily for discrete ones. Finally, $I_{ccs}$ [36] can be applied to discrete or continuous data, regardless of the distribution (although it can result in counter-intuitive results in some contexts). When using binary, spiking data, the relevant redundancy function will be different from when attempting to analyze largely Gaussian fMRI-BOLD data. There are currently no well-used redundancy functions for the case of non-normal, continuous data (such as LFP signals), and in these cases, some compromises must be made (either by using Gaussian methods or by discretizing the data). If one chooses to discretize the data, we recommend keeping the number of bins low (ideally, binarizing the data) in order to avoid finite-size effects and the explosion of the joint probability space (discussed below).

In addition to concerns around data type, there are also practical concerns about the size of the available dataset that can be used to infer joint probability distributions. For a system of *k* interacting variables, any multivariate information-theoretic analysis (beyond just the PID) requires a *k*-dimensional joint probability distribution to be constructed, which can require incredibly large amounts of data as *k* becomes large. For a system of binary elements such as spiking neurons, there are $2^{k}$ possible combinations of 0s and 1s. When analyzing recordings with hundreds or thousands of neurons (as might be recorded on a multielectrode array or a Neuropixels probe), care should be taken to ensure that there are enough data for it to be possible to robustly infer the probabilities of all configurations, as undersampling and finite-size effects can severely compromise the estimated entropies and mutual information data. This is one reason why it is generally considered inadvisable to discretize continuous data by binning it into a large number of communities [15], as pipelines that use large numbers of bins almost certainly do not have sufficient data to brute force the full space, and the resulting inferences will be severely compromised. There are, however, practical guides to handling sampling in real-world empirical settings [59,60].

### 6.1. Tools for Data Analysis

Since the original proposal by Williams and Beer [1], a number of different scientific programming packages have been released with implementations of the various redundancy functions. The most comprehensive is the Discrete Information Theory (DIT) toolbox [61], which provides functions for a large number of different redundancy functions and an automated PI-lattice solver that can solve lattices for two to five sources and a single target. DIT is written to operate on joint probability distributions; consequently, if one wishes to use DIT for analysis of time series, the inference of the multivariate distribution must be performed “by hand” before being passed to DIT. DIT also includes a large number of other information-theoretic measures used in computational neuroscience, including the total correlation [62], the Tononi–Sporns–Edelman complexity [63], and various mutual information measures. DIT is available only in Python.

For researchers interested in operating directly on time series, the IDTxl package [54], provides a Python-language framework for information-theoretic analyses of time series. IDTxl has two built-in PID solvers: one using the $I_{sx}$ redundancy function [40] and the other using the $I_{BROJA}$ measure [24]. IDTxl also includes a large number of algorithms for bivariate and multivariate transfer entropy inference (based on the JIDT package [64]). The $I_{sx}$ function has also been released in a stand-alone, Python-language package called SxPID [40]. We include SxPID in particular, because it is currently the only package that can support time-resolved, local measures in addition to expected ones. Like DIT, SxPID operates on joint probability distributions rather than time series, and therefore these distributions must be constructed.

Finally, for MATLAB users, there is the Neuroscience Information Theory Toolbox [9]. This package only supports the $I_{\min}$ redundancy function, and like IDTxl it also includes functions for transfer entropy inference, although it lacks the additional algorithms and multivariate extensions that have been developed since its release.

### 6.2. Interpreting PID Results as Neural Information Processing

One important caveat to note with respect to PID applied to neural data is that while it provides a powerful framework for recognizing statistical dependencies between neurons, it does not necessarily provide a *mechanistic* explanation for *why* particular neurons are synergy-dominated or redundancy-dominated. Now that the existence of statistical synergies is well replicated, a natural future avenue of research might be to attempt to determine how particular biological aspects of neuronal function (e.g., neurotransmitter systems, neural architecture, etc.) produce redundant, unique, or synergistic computational dynamics.

### 6.3. Practical Limitations of the PID Framework

The PID framework has a number of limitations that potentially complicate its application in naturalistic settings. The most significant is the explosive growth of the PI lattice. For a system with *k* parent neurons, the number of distinct PI atoms is given by the *k*th Dedekind number (minus two) [30]. This sequence of numbers grows appallingly fast: in the case of six parent neurons, there are 7,828,354 distinct PI atoms, the vast majority of which are difficult to interpret. Dedekind numbers greater than nine are currently unknown. Given that single neurons can receive inputs from very large numbers of upstream neurons, a “complete” description of the information dynamics of any individual neuron is completely intractable. This issue may be partially addressed by the development of heuristic measures of redundancy and synergy such as the O-information [65], which has been applied to information dynamics in networks of neurons [66] and human functional magnetic resonance imaging blood-oxygen-level-dependent (fMRI-BOLD) signals [67], although these measures typically trade completeness for scalability.

PID also inherits a limitation common to most applied information-theoretic frameworks, i.e., the problem of accurately estimating entropies and mutual information. When naively estimating probabilities from a finite dataset, it is known that the plug-in estimator for the Shannon entropy consistently *underestimates* the true entropy and *overestimates* the mutual information (for example, two processes that are uncorrelated can have non-zero apparent mutual information) [68]. While this is less of an immediate concern in the case of single-neuron recordings, where the state space is small (binary) and recording times are long (often of the order of millions of samples), this concern can become an issue as the state space grows. Consequently, great care should be taken when performing a PID (or any other information-theoretic) analysis to ensure appropriate null-model comparisons and that the limitations of the chosen entropy estimator are understood (for a more detailed discussion of information estimation in finite-sized datasets, see [15]).

## 7. Future Directions

Research on the foundations of multivariate information theory and its application to complex systems is ongoing and forms an active field in its own right. In this review, we have largely focused on “classic” PID, as proposed by Williams and Beer [1], due to its widespread application in neuroscience. There are, however, a number of later advances that extend the PID framework in potentially fruitful directions. While detailed discussion of these avenues is beyond the scope of this article, we will briefly mention them here and provide references for interested readers.

### 7.1. Local PID

So far, we have focused exclusively on decomposing the “expected” mutual information into partial information atoms. Measures such as the redundancy, synergy, etc. are calculated over the entire probability distribution of states that the parents and target can adopt. While this is standard practice in most applications of information theory, the PID framework itself can also be localized to single configurations. Consider the definition of joint mutual information given in Equation (Equation 3). It can be seen that this is an expected value:(12)$$I\left(X;Y\right)=E_{X,Y}\left[\log_{2}\frac{P(y|x)}{P(y)}\right]$$

We then define the *local mutual information* associated with the particular configurations **X** = **x** and Y=y as:(13)$$i\left(x;y\right)=\log_{2}\frac{P(y|x)}{P(y)}$$

The local PID framework allows us to perform the same decomposition, but for every individual possible configuration of specific source and target states. In the context of neural recordings, the obvious application of this is the construction of a time-resolved, “informational time series”, where the particular redundant, synergistic, and unique modes of information sharing can be resolved for every moment in time. For more on local information theory and its applications to complex systems, see [41]. This kind of time-resolved analysis has already begun to bear fruit, for example in finding that neuronal avalanches (a common signature of complex, critical dynamics in the nervous system) have distinct “computational profiles”, with the levels of various partial information atoms varying over the course of a single avalanche [45].

For a local PID analysis to be possible, all that is required is that the redundancy function chosen is localizable. The original measure $I_{\min}$, proposed by Williams and Beer [1], did not satisfy this property, and neither did many subsequent refinements such as the measures developed in [24,25,31]. To date, three local redundancy functions have been proposed: $I_{cs}$ by Ince [36], $I_{\pm }$ by Finn and Lizier [37], and $I_{sx}$ by Makkeh et al. [40] (a subsequent elaboration on $I_{sx}$, termed $I_{\tau sx}$, is described below).

### 7.2. Multi-Target and Temporal PID

The original PID framework allows for the decomposition of the information that an arbitrary set of parents provides about a single target. A natural generalization is to allow for multiple targets. For example, what synergistic information about two targets is disclosed by the synergistic joint state of two parents? In the context of a temporally evolving process, the same analysis could be used to decompose the redundant, unique, and synergistic ways that the *past* constrains the *future*. A recent multi-target generalization of the PID, termed *integrated information decomposition* (ΦID) [69,70] has been proposed, which opens the door to a much wider range of possible analyses than is possible with a single-target PID analysis. ΦID analysis of fMRI data has found changes in the flow of synergistic information associated with loss of consciousness [46,71] and the genetic architecture of the brain [44]. A recently introduced “temporal redundancy function” ($I_{\tau sx}$), coupled with a ΦID analysis, led to the discovery of a set of sixteen distinct information dynamics associated with the temporal evolution of pairs of interacting neurons, many of which were previously unknown [45]. Due to its comparative novelty, multi-target information decomposition has received less attention than its single-target counterpart, and consequently, a number of open questions remain.

## 8. Summary

In this paper, we have aimed to provide an accessible introduction to the partial information decomposition framework [1] and shown how it can be applied to answer fundamental questions about information processing in neural circuits. We have focused particularly on statistical synergy, i.e., the information about the future of a target neuron that is disclosed by the higher-order patterns instantiated by multiple upstream neurons and is irreducible to any single source, as a measure of processing or computation. The existence of synergy in empirical data shows us that neurons do not appear to blindly sum the number of inputs and fire in response to the total (as would be expected from a basic threshold model or complex contagion model), but instead they are also sensitive to the particular patterns of incoming stimuli. Furthermore, not only does the existence of statistical synergy add nuance and dimensions to our understanding of the dynamics of individual neurons, we have also shown that the patterns of synergistic information processing are informed by the local environment in which those neurons are embedded (rich club membership, motifs, clustering, etc.), as well as the behavioral state of the oganism under study. These findings hint at the existence of a large, potentially fruitful space of future research relating higher-order information dynamics to biological, cognitive, systemic phenomena at many scales of analysis.

## Figures and Tables

**Figure 1 entropy-24-00930-f001:**
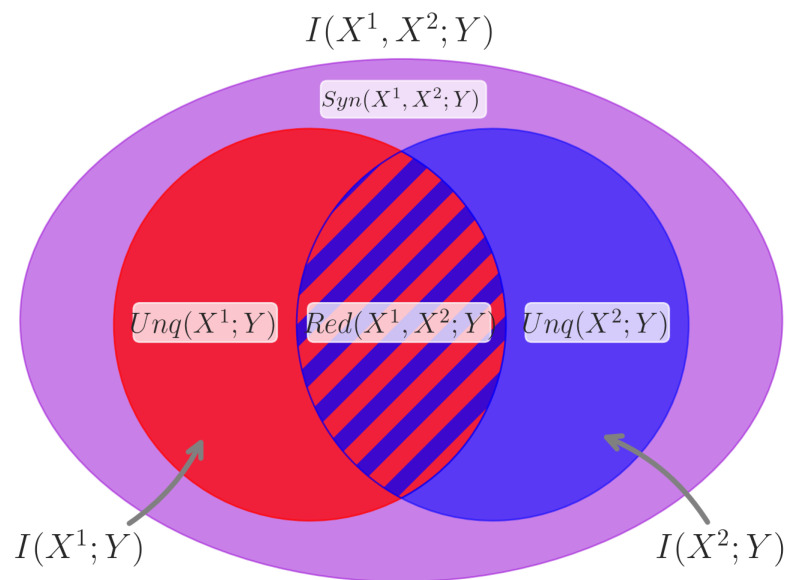
The total information contained in the activity of two source neurons $X^{1}$ and $X^{2}$ about the activity of a target neuron *Y* consists of multiple parts. This total information, $I(X^{1},X^{2};Y)$, is represented by the outermost oval. Contained within this total is the information that $X^{1}$ and $X^{2}$ each independently carry about Y, represented by the red circle for $I(X^{1};Y)$ and the blue circle for $I(X^{2};Y)$. These independent sources can carry redundant information, represented by the overlapping striped section labeled $\mathrm{Red}(X^{1},X^{2};Y)$. The non-redundant information each source neuron accounts for is the unique information $\mathrm{Unq}(X^{1};Y)$ and $\mathrm{Unq}(X^{2};Y)$. Finally, and most relevantly for the study of information processing, the *joint state* of $X^{1}$ and $X^{2}$ can account for the activity of Y to some degree. This information is not accounted for by either source independently and is the synergistic information that $X^{1}$ and $X^{2}$ carry about *Y*, i.e., $\mathrm{Syn}(X^{1},X^{2};Y)$. The purple space making up the difference between what is included in the $I(X^{1};Y)$ and $I(X^{2};Y)$ circles and the total information $I(X^{1},X^{2};Y)$ represents $\mathrm{Syn}(X^{1},X^{2};Y)$.

**Figure 2 entropy-24-00930-f002:**
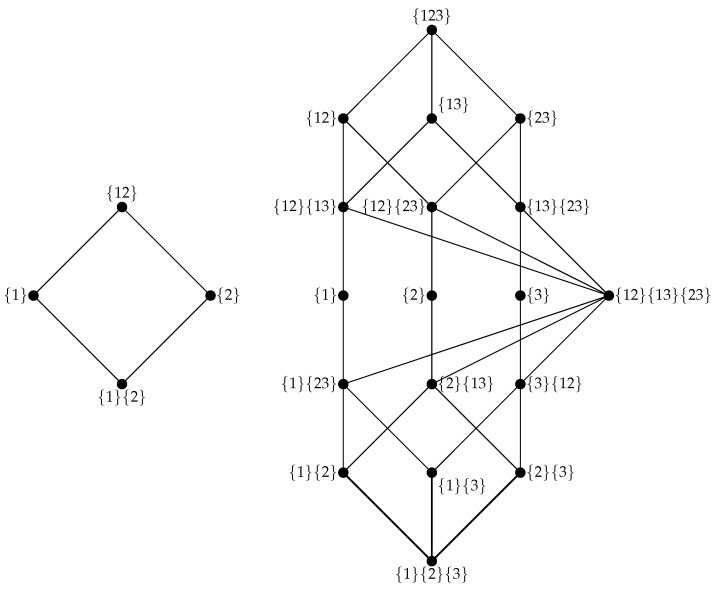
Partial information lattices. On the left is the lattice for two predictor variables, and on the right is the lattice for three predictor variables. Each lattice is constructed and annotated following the notation in [1]. Lattice vertices are partial information atoms, i.e., the unique modes of information-sharing that comprise the overall joint mutual information. Information atoms are denoted by index only: for example, {1}{2} is the information redundantly disclosed by $X^{1}$ or $X^{2}$, {1}{23} is the information disclosed by $X^{1}$ or ($X^{2}$ and $X^{3}$), etc. Lattice edges indicate which atoms subsume other atoms. Atoms connected to and below other atoms consist of components of and/or subsets of the higher atoms, for example, {1}{2}⪯{1}{23} and {2}{13}, since information disclosed by {1}{2} would also be visible to {1}{23} and {2}{13} if we did not use the Mobius inversion.

**Figure 3 entropy-24-00930-f003:**
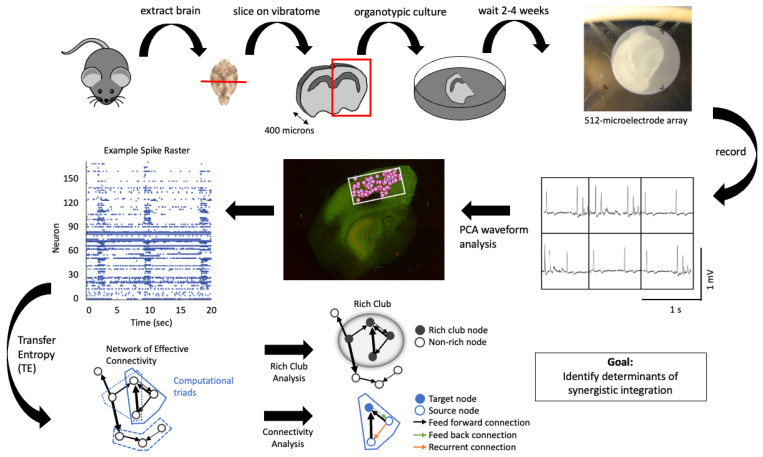
Example of how neuronal activity recordings can be subjected to PID-based study. Rapid extraction and sectioning of mouse brains created 400 μm sagittal slices of cortex. Incubation in nutrient media allowed the slice to culture, restoring organotypic connectivity patterns [47]. Placing the culture on a high-density 512-channel microelectrode array allowed for recordings permitting spike sorting via PCA waveform analysis, to attribute observed spikes to individual neurons. The resulting spike rasters reflect the millisecond-level precision activation patterns of hundreds of neurons. Analyzing the effective connectivity among neurons using transfer entropy enables extraction of the full effective network upon which the spiking dynamics occurred. The computational triads, consisting of two neurons (source neurons) connecting to a common third neuron (target neuron), can then be identified. PID can then be applied separately to each computational triad. Variations across triads can be used to analyze how synergy (or redundancy) varies as a function of network properties such as the boundaries of a rich club or as a function of the number of feedback and recurrent connections. Figure adapted from [2].

**Figure 4 entropy-24-00930-f004:**
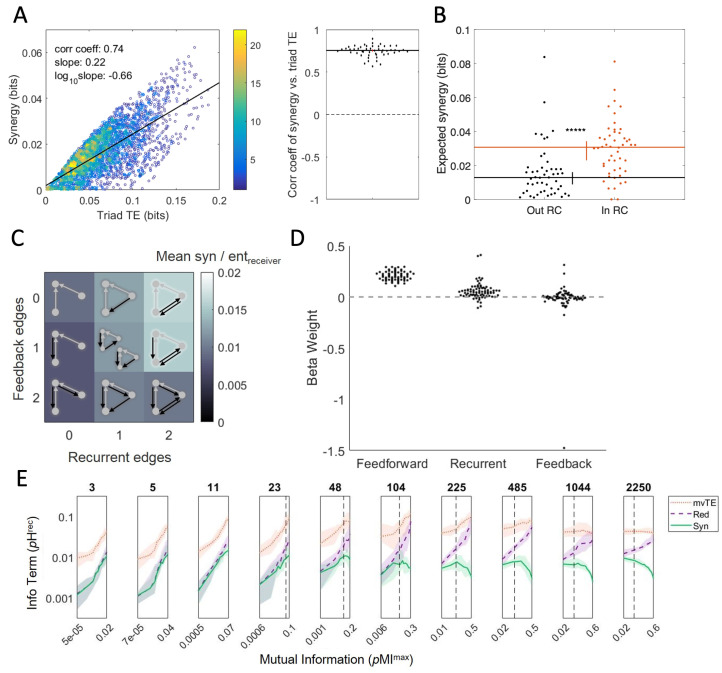
Empirical study of circuit dynamics using PID can reveal covariates of synergistic integration. (**A**) Synergistic integration is robustly positively correlated with the strength of feedforward connections in a triad. **Left**: A single representative network with 3000+ computational triads shows that feedforward connection strength (i.e., triad TE) was positively related to the synergy for each triad. **Right**: This positive relationship was reliably observed over 75 different effective networks analyzed from 25 different cultures. Images from [2]. (**B**) The mean synergy across all triads localized to rich clubs was significantly higher than the synergy in triads not localized to a rich club. The inset ‘∗∗∗∗∗’ indicates $p<1\times 10^{-9}$ for the difference between conditions. Image from [2]. (**C**) Sorting computational triads based on the number of connections from the target neuron back to the source neurons (feedback connections) and connections between the source neurons (recurrent connections) revealed more synergy for triads with more recurrent connections. Image from [4]. (**D**) Feedforward and recurrent connections were both positively related to synergistic integration, while feedback connections were not reliably related to synergistic integration. Image from [4]. (**E**) Synergy (normalized to be the proportion of the receiver entropy, or *p*H${}^{rec}$) was non-monotonically related to the similarity of the spiking of the source neurons (normalized to be the proportion of the maximum possible mutual information, or *p*MI${}^{max}$) when analyzed across a wide range of timescales. Each panel shows the results for a single timescale. Note that the x-axis is logarithmically scaled, and the range varies across panels. The timescale varies from left to right across panels, ranging from short (e.g., 3 ms, 5 ms, and 11 ms) to long (e.g., 485 ms, 1044 ms, and 2250 ms), as indicated above each panel. At short timescales, the maximum similarity in spiking between source neurons was low, and both synergy (green solid line) and redundancy (purple dashed line) were positively related to the similarity of the spiking between source neurons. At long timescales, the total similarity was high, and only redundancy was positively related to the similarity of the source neurons. Synergy was maximized when the source neurons were intermediately similar, with *p*MI${}^{max}=\;0.07$, as marked with a vertical dashed line in each panel. Image related to the similarity from [3].

## Data Availability

Not applicable.

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
