# Peer review of "Revealing the Dynamics of Neural Information Processing with Multivariate Information Decomposition"

_entropy, 2022, doi:10.3390/e24070930_

Round 1
Reviewer 1 Report
Dear Authors,
The paper reads well and presents a nice introduction and overview of the multivariate information decomposition and neural activity analysis research fileds. However, I suggest some improvements:
- page 2: write out the abbreviation LFP;
- page 6, Figure 2: it is not clear from the graph how the edges are connected with the vertices; a description of the graph construction process in the paper would be beneficial;
- page 6, Figure 2 caption: change $X_3$ to $X^3$;
- page 7, Figure 3: in my opinion, the figure is too similar to Figure 1 in [2], please make considerable changes;
- page 8, Figure 4: sub-figures A, B, C and D are the same as in Article [4], please redesign them;
- page 8, Figure 4: variables $mvTE$, $ent_{receiver}$, $p$, $pH^{rec}$, $pMI^{max}$ appear in the image and in the caption for the first time; please introduce them in the text or follow the same convention. for example, using Syn(..) for synergy throughout the paper.
Best Regard,
Reviewer
Author Response
We have pasted the full set of reviewer comments here in bold. Our responses are inset in regular font.
Dear Authors,
The paper reads well and presents a nice introduction and overview of the multivariate information decomposition and neural activity analysis research fileds. However, I suggest some improvements:
page 2: write out the abbreviation LFP;
- Done. Thanks!
- page 6, Figure 2: it is not clear from the graph how the edges are connected with the vertices; a description of the graph construction process in the paper would be beneficial;
-
Thank you for highlighting this omission. The main text itself already indicated this but we added a reference to the figure to emphasize this point. We have revised the caption to clarify. The new caption reads as follows (red marks the edits):
- "\caption{\textbf{Partial information lattices.} On the left is the lattice for two predictor variables, and on the right is the lattice for three predictor variables. \color{red}Each lattice is constructed and annotated \color{black} following the notation in \cite{williams_nonnegative_2010}. \color{red} Lattice vertices are partial information atoms: the unique modes of information-sharing that comprise the overall joint mutual information.\color{black} Information atoms are denoted just by index: for example $\{1\}\{2\}$ is the information redundantly disclosed by $X^1$ or $X^2$, $\{1\}\{23\}$ is the information disclosed by $X^1$ or ($X^2$ and \color{red}$X^3$\color{black}), etc. \color{red}Lattice edges indicate which atoms subsume other atoms. Atoms connected to and below other atoms consist of components of and/or subsets of the higher atoms, for example $\{1\}\{2\} \preceq \{1\}\{23\}$ and $\{2\}\{13\}$ since information disclosed by $\{1\}\{2\}$ would also be visible to $\{1\}\{23\}$ and $\{2\}\{13\}$ if we did not use the Mobius inversion.\color{black}}"
page 6, Figure 2 caption: change $X_3$ to $X^3$;
- Done. Thanks!
page 7, Figure 3: in my opinion, the figure is too similar to Figure 1 in [2], please make considerable changes;
- This is a reproduction of our figure as cited. Unless deemed editorially necessary, we strongly prefer to leave the figure unaltered to maintain transparency.
page 8, Figure 4: sub-figures A, B, C and D are the same as in Article [4], please redesign them;
- These are reproductions of our figures as cited. Unless deemed editorially necessary, we strongly prefer to leave the figures unaltered to maintain transparency.
page 8, Figure 4: variables $mvTE$, $ent_{receiver}$, $p$, $pH^{rec}$, $pMI^{max}$ appear in the image and in the caption for the first time; please introduce them in the text or follow the same convention. for example, using Syn(..) for synergy throughout the paper.
-
Thanks. We have updated the caption to describe these terms in the same language as used in the main text.
Best Regard,
Reviewer
Reviewer 2 Report
I would like to recommend the paper for publication.
Some small comments,
- Probably the '****' at Fig 4 was unintended, if so, please fix.
- It would have been nice, though not sure if possible in this set-up, to see more analysis on synergy via O-information. Although it is briefly mentioned near the end, inspecting O-information as a function of the maximum interaction order as in (arxiv:1902.11239) would be very telling on the contribution of different architectures.
Author Response
The full review has been copied here in bold. Responses are inset and regular font.
I would like to recommend the paper for publication.
Some small comments,
Probably the '****' at Fig 4 was unintended, if so, please fix.
-
These asterisks were intended, but clearly were confusing. To resolve, we updated the surrounding text to read:
-
“The inset '$*****$' indicates $p < 1\times10^-9$ difference between conditions.”
-
It would have been nice, though not sure if possible in this set-up, to see more analysis on synergy via O-information. Although it is briefly mentioned near the end, inspecting O-information as a function of the maximum interaction order as in (arxiv:1902.11239) would be very telling on the contribution of different architectures.
- We agree that O-information is an important new development. Indeed, we already cite the arxiv paper indicated (Rosas et al., 2019). As noted, it is difficult in this set-up to unpack much more substantially. We have intentionally kept the unpacking of individual approaches minimal and rather pointed readers towards the most relevant sources. We have, nonetheless, added another reference in this section in recognition of the reviewer's suggestion. The relevant section now reads:
- "This issue may be partially addressed by the development of heuristic measures of redundancy and synergy, such as the O-information \cite{rosas_quantifying_2019}, which has been applied to information dynamics in networks of neurons \cite{stramaglia_quantifying_2021} and human functional magnetic resonance imaging blood oxygen level devepedent (fMRI-BOLD) signal \cite{Varley_OInformation_2022}, although these measures typically trade completeness for scalability."